# A New Composite Fractal Function and Its Application in Image Encryption

**DOI:** 10.3390/jimaging6070070

**Published:** 2020-07-15

**Authors:** Shafali Agarwal

**Affiliations:** Independent Researcher, 9600 Coit Road, Plano, TX 75025, USA; shafali.agarwal@gmail.com; Tel.: +1-916-693-4645

**Keywords:** composite fractal function, henon map, z-scan, random fractal matrix, permutation, diffusion

## Abstract

Fractal’s spatially nonuniform phenomena and chaotic nature highlight the function utilization in fractal cryptographic applications. This paper proposes a new composite fractal function (CFF) that combines two different Mandelbrot set (MS) functions with one control parameter. The CFF simulation results demonstrate that the given map has high initial value sensitivity, complex structure, wider chaotic region, and more complicated dynamical behavior. By considering the chaotic properties of a fractal, an image encryption algorithm using a fractal-based pixel permutation and substitution is proposed. The process starts by scrambling the plain image pixel positions using the Henon map so that an intruder fails to obtain the original image even after deducing the standard confusion-diffusion process. The permutation phase uses a Z-scanned random fractal matrix to shuffle the scrambled image pixel. Further, two different fractal sequences of complex numbers are generated using the same function i.e. CFF. The complex sequences are thus modified to a double datatype matrix and used to diffuse the scrambled pixels in a row-wise and column-wise manner, separately. Security and performance analysis results confirm the reliability, high-security level, and robustness of the proposed algorithm against various attacks, including brute-force attack, known/chosen-plaintext attack, differential attack, and occlusion attack.

## 1. Introduction

A new digital world has increased the transmission demand of multimedia content at large scale. At the same time, excessive use of digital media such as social networking sites and instant messaging brings about the risk of invading to the personal privacy of its owner. Sometimes, this is required to be more conscious about the secure receiving of image contents like in military security documents, medical diagnostic images, bank account-related information from financial institutions, government offices, etc. Thus, it is of the utmost concern to protect the system from the serious risk of confidential data leakage [1].

The image encryption technique was introduced to secure the images transmitted through the smartphone, IPAD, computer, laptop, and other electronic devices over the internet or outsourced to the cloud storage. As compared to text and binary data, image encryption is more critical due to its additional characteristics such as highly correlated adjacent pixels in different directions, excessive redundancy, and greater data volume. Although many conventional encryption algorithms such as Data Encryption Standard [2], RSA [3], Advanced Encryption Standard [4], and International Data Encryption Algorithm [5] with high data transmission security have been proposed. However, these ciphers proved inappropriate and time consuming for image information encryption. The foremost concern of implementing image-specific cryptosystem to resolute image information protection includes the consideration of all necessary image characteristics. Over the years, various cryptosystem design techniques have been developed, comprising a chaotic system, DNA structure, fractal function, wavelet transform, cellular automata [6,7,8,9,10,11], and many more.

The author in [12] suggested that a cryptosystem having two fundamental structures, i.e. confusion-diffusion, exhibits resilient performance against cyber-attacks. The confusion phase deals with the pixel shuffling or pixel permutation to reduce the relationship between the adjacent image pixels. Whereas pixel diffusion ensures that the small change in plain image information should affect at least half of the cipher image information. Pixel shuffling is achieved by changing the pixel positions within the image, and the pixel diffusion is performed by changing the pixel values. Both phases can be repeated as many times as needed to attain the desired security level. However, it costs the total execution time.

Over the last decade, many image encryption ideas have been proposed using the same structure [10,13,14,15,16,17,18]. An image encryption algorithm can perform permutation in two ways: (1). Bit-level permutation [19,20,21], and (2). Pixel-based permutation [22,23,24]. A bit is considered as the smallest operating element. To perform a bit-level permutation, an image needs to be transformed into the binary form. A bit-plane scrambling shuffles the bits itself, and hence, is considered more secure. However, it requires more execution time compared to the pixel-plane scrambling. For example, the author in [25] proposed a bit-level permutation by generating a new sequence dynamically on every small change in the plain image to ensure the robustness of the system against the chosen-plaintext and known-plaintext attacks. Shouliang Li et al. pointed out that by implementing bit-plane and pixel-plane scrambling jointly can have better performance in terms of security as well as a speedy outcome [26]. A bit-plane scrambling plus diffusion (SPD) operation was proposed based on the card trick method to change the pixel position and value at the same time [14]. Before performing it, a pixel-level scrambling is also done to randomize the plain image pixels. Another example of simultaneous pixel confusion and diffusion was proposed in [13]. Both processes were implemented through horizontal confusion-diffusion followed by vertical confusion-diffusion with the help of a newly proposed chaotic map, i.e. tent delay-sine cascade with the logistic map.

A high-efficient pixel scrambling was implemented in a block-based square matrix. The method used a chaotic sequence to shuffle the pixels among the different blocks, following which the pixel positions were changed again within the block itself using another chaotic sequence [23]. A simple yet difficult to breach cryptosystem was designed by the author [24], in which pixel-based scrambling was implemented using the Arnold method and Lucas series. A total of nine rounds of pixel permutation were executed to enhance the efficiency and uniformity of the pixels. Xuncai Zhang et al. came up with a concept of local pixel scrambling and global pixel scrambling, which was executed multiple times using a Hilbert curve and piecewise linear chaotic map (PWLCM) respectively [22]. The overall encryption process performed by executing multiple rounds of pixel scrambling and pixel diffusion alternatively. A two-pass method of pixel permutation introduces the utilization of the first half-pixel values to permute the other second half and vice-versa [27]. The author proposed a simple and fast one-round-based dynamic diffusion operation in which the sum of the plain image pixels plays an important role to have an extremely sensitive and secure cryptosystem. At each pixel diffusion step, the sum of the pixels will be updated to significantly increase the resistivity against the known-plaintext, chosen-plaintext, and differential attacks [28]. Recently, a two-steps diffusion process was proposed in [29]. In the method, a new two-dimensional chaotic infinite collapse map generates two matrices X and Y to perform pixel substitution. In the first step, scrambled image pixel values were rearranged according to the sorting index generated through a one-dimensional chaotic sequence X. Further, perform the diffusion operation on a previously generated sequence using another chaotic matrix Y.

A fractal is a self-similar image that is infinitely scalable in all directions and encompasses striking features to study and explore. The fractal function works in a complex domain and generates complex patterns using complex numbers upon execution. The chaotic nature of the fractal function exhibits extremely random behavior and excellent sensitivity towards its initial condition, which makes eavesdroppers inefficient to replicate the secret key to retrieve the confidential information [30,31,32,33]. Moreover, a complicated structure of a fractal function leads to a reliable cryptosystem design that is capable of secure digital information transmission. A fractal possesses a small footprint, which means only a few parameters are required to produce a unique fractal image. The image instantly reacts to even a tiny changed input, resulting in a completely different fractal image. At an early stage, the USA navy published a patent describing the use of fractal function as a secret key in an image encryption method [34]. However, a fractal function can be used in any phase of a cryptosystem design, i.e. key design or encryption/decryption. The author utilized a superior fractal function to generate three initial values, which were further inputted to a newly proposed chaotic map to produce three secret key sequences in [35]. The Mandelbrot set, a most famous fractal function introduced by B.B. Mandelbrot in 1979 [33], has a wide exploration history due to its interesting and complex structure. In the last decade, many image encryption methods have been proposed using the Mandelbrot set and a Julia set [36,37,38,39,40]. A set of four Julia images was used in the secret key design, which was further modified by the Chen hyper-chaotic sequence. The same chaotic sequence was also used to select a different fractal image to be further applied at each diffusion phase [36]. The real values of a Julia set function have applied to substitute the scrambled pixels by multiplying the values together [37]. The author suggested an image encryption method in which an H-fractal was utilized to diffuse the scrambled image pixels [22,41]. The author in [42] proposed a fractal function-based cryptosystem, in which three sets of initial values are derived by iterating a conjugate transcendental fractal function using Picard, Mann, and Ishikawa iteration methods. An image encryption algorithm was proposed by utilizing the RSA algorithm elements with the fractal quadratic transformation system, best suited to the images that allow clearly distinguishing the contours [40]. A hybrid method including fractal-image addition along with binary encoding has been proposed to encrypt the digital images [39]. A finite field cosine transformation-based multiphase symmetry key encryption algorithm was designed by utilizing the fractal as a source of randomness to generate a one-time pad keystream [43]. A burning ship fractal function was employed to generate a secret key sequence, which uses Hilbert transformation to enhance the sequence randomness [44].

A generalized M-J set was introduced to generate a secret key by using environmental noise and the SHA-512 method. The indefinite nature of environmental noise ensures key security at every use. The author also succeded to achieve a quite large keyspace i.e. 2^3250^ [45]. The chaotic nature of inverse fractal interpolation function was used to generate a pseudo-random key sequence, which was further utilized in plain image pixel scrambling [46]. A symmetric key encryption algorithm was proposed by the author in which an encryption key was generated using the Mandelbrot set functions [47]. Later the author in [48] pointed out that the method is not resistant to the chosen-plaintext attack, chosen-ciphertext attack, and known-plaintext attack, under the assumption of direct use of the plain image. Similarly, cryptanalysis was carried out on secure fractal image coding based on fractal parameter encryption method. The basic idea of the paper is to encrypt few fractal parameters using fractal encoding [49], but the intruder can reconstruct the image by comparing multiple cipher images of the same original image encrypted using different secret keys [50]. By analyzing the above-mentioned algorithms, and considering the chaotic nature of a fractal function, it is inferred that a non-linear iterative fractal function is a well-suited structure of a cryptosystem design. Though many researchers have already applied fractal theory in the image encryption method, it was mostly aligned with other elements such as a chaotic map.

In this study, a new composite fractal function (CFF) is proposed which is a combined form of two different standard Mandelbrot set functions to achieve high sensitivity and strong security. The resulting CFF has better chaotic complexity in terms of trajectory in large space, a self-similar structure at each scale, fractional dimension, and positive Lyapunov exponent. The main idea of this paper is to utilize the same function in pixel confusion as well as in the pixel diffusion process using different fractal-based key sequences that are generated by varying the control parameter value.

The rest of the paper is organized in the following sections: In Section 2, the structure of newly proposed CFF along with its dynamic characteristics is discussed. Section 3 focuses on the proposed image encryption algorithm. The simulation result and performance analysis are discussed in Section 4. At last, the paper is concluded.

## 2. A New Composite Fractal Function (CFF)

A Mandelbrot fractal function was invented by B.B. Mandelbrot in 1982 [33] while analyzing the topology of the famous Julia set function. According to him, a simple mathematical function can create the theory of self-similarity and roughness by generating the complex fractal images. The Mandelbrot set (MS) can be defined as a set of c values in the complex plane for which the orbit of 0 remains bounded under iteration of the given function.
(1)$$z_{n+1}=z_{n}^{2}+c$$

The other modified form of the MS function can be defined as:(2)$$z_{n+1}=z_{n}^{4}+c$$

Figure 1 shows the fractal images generated by implementing both functions in MATLAB.

The below section demonstrates the proposed composite fractal function (CFF) by combining the discussed Mandelbrot set function and its modified version.The characteristics of the function are investigated using the parameters given in Section 2.2.

### 2.1. The Mathematical Definition of CFF

A combined form of a Mandelbrot function and its variant, including an additional parameter, i.e. β enhances the nonlinear dynamics of the MS function. Mathematically, the CFF is defined by Equation (1).
(3)$$z_{n+1}=z_{n}^{2}+c+\beta *\left(z_{n}^{4}+c\right)$$
where c∈C and 0<β<1 with $z_{0}=0$. The fractal image is the set of complex points for which the orbit of $z_{n}$ does not tend to infinity while n approaches to infinity. The paper uses three chaotic sequences that are generated by varying the parameter β value. Figure 2 shows the fractal images generated from Equation (3) by putting β = 0.2, β = 0.3, and β = 0.5.

### 2.2. Dynamical Properties of the Proposed CFF

The chaotic properties of the proposed fractal function are evaluated using a self-similar structure, fractal dimension, trajectory, and Lyapunov exponent.

#### 2.2.1. Self-Similar Structure of CFF

The self-similarity feature displays the look-alike object upon scaling the object as the whole or a part. A Mandelbrot set image falls under the category of escape time fractal. It means that the image can be unequally scaled in different directions. The same mechanism of scaling has been applied here because the Mandelbrot set is used as a seed map to construct the proposed CFF structure. The implementation result shows the self-similar structure displayed by both CFF functions on various scaling factors. Few of them are shown in Figure 3.

#### 2.2.2. Fractal Dimension

A fractal function is known to have a fractional dimension because of its complex fractal geometry. The concept of the ’in-between’ dimension was introduced by a mathematician Felix Hausdorff in 1918 [51]. According to him, a quantitative measurement of perimeter roughness is calculated as a fractal dimension. Since in the classical geometric world, the dimension of a smooth line is one, a plane has a dimension of two and so on, which implies that the Hausdorff dimension of a two-dimensional irregular fractal image must lie between one and two.

Here the fractional dimension is calculated using a box-counting method. Recently, Murat Erhan Çimen et al. developed a user-friendly interface to calculate the fractal dimension using a box-counting method [52]. Mathematically, the process starts by superimposing the complex and irregular fractal image with the grid of squares N over the edge. If the number of squares occupied by the fractal image edge is E, then the formula to calculate the dimension is given as:(4)$$dim=\frac{\log N}{\log E}$$

According to the above-discussed method, the fractal dimensions of the proposed CFF for β = 0.2, 0.3, and 0.5 have the values 1.3479, 1.3197, and 1.2776, respectively.

#### 2.2.3. Trajectory

The trajectory of a function is obtained by evaluating the mapping orbit for a set of initial values. This is a visual method to analyze the randomness and ergodicity of a chaotic map. For better chaotic performance, the outcome of each iterated point should occupy the large phase space and also possesses a complicated geometrical structure. To demonstrate the orbit structure of the proposed CFF, the function is iterated 1600 times for each β value and the values are plotted in a 2D phase space. From Figure 4, it can be seen that the given CFF exhibits complex behavior in all parameter ranges, and its trajectory is distributed in the entire phase space. According to the trajectory diagram of MS function shown in Figure 4a, it can be concluded that the chaotic range of CFF is expanded in larger phase space as compared to the MS function. The visual representation of function outcome indicates that the proposed fractal function has strong randomness and ergodicity property.

#### 2.2.4. Lyapunov Exponent

A sensitiveness property of the chaotic system exhibits two exponentially diverted trajectories of an extremely closed initial point over time. It indicates that the system’s orbit is unpredictable. For a function f(x), the feature is quantitatively estimated using the Lyapunov exponent. The formula can be defined as:(5)$$LE_{f\left(x\right)}=\underset{n\to \infty }{\lim}\left\{\frac{1}{n}\ln\left|\frac{f^{n}\left(x_{0}+\varepsilon \right)-f^{n}\left(x_{0}\right)}{\varepsilon }\right|\right\}$$
where ε denotes the difference between two initial points to be considered. A positive LE value indicates that the orbits of two initial points will be completely separated over the successive iterations. It proved that the system is in a chaotic state. A system having more than two positive Lyapunov exponent values is considered as a hyperchaotic map.

The Lyapunov exponents of MS and CFF are calculated for both β values, i.e. 0.3 and 0.5, and plotted in Figure 5a–c, respectively. From these diagrams, it is evident that as compared to MS function, the proposed function has high Lyapunov exponent values in a large continuous interval. It means that the system has a wider chaotic range with no periodic window. The largest LE value of MS is 1.44 whereas the CFF has a maximum LE value close to 5.89 in both cases. Thus, the result signifies the strong, unpredictable behavior of the new fractal data sequence.

## 3. A Cryptosystem Design Using Composite Fractal Function

The paper focuses on designing image encryption & decryption method using a new proposed CFF. An important point about the role of fractal image in the cryptosystem design includes its utilization at the various stages. The method executes the CFF three times to extract three different fractal data sequences used to confuse and diffuse the plain image, respectively. To achieve a higher level of security and computation efficiency, two rounds of pixel scrambling and substitution are implemented to generate a noise-like cipher image. Figure 6 shows the block diagram of the proposed image encryption method.

### 3.1. Secret Key Sequence Generation

An integral component of an image encryption algorithm is the secret key used to encrypt and decrypt the image that is supposed to be strong enough to resist the Brute-force attack. A secret key is derived by iterating the fractal function and has a direct impact on pixel shuffling and substitution processes. Therefore, the randomness and chaotic behavior of the key sequence must be complex enough to provide a highly secure cryptosystem. The proposed method uses three different fractal-based key sequences. The first one-dimensional key stream is used to generate a random two-dimensional matrix, which will be further applied in the plain image pixel scrambling process. In the process to generate a one-dimensional fractal data vector, two steps need to be implemented.(1)Generate a fractal data matrix F1 using CFF by considering β=0.2 given in Equation (3). For example, if the method is encrypting an image of size 256 × 256, then execute the CFF to generate a 100 × 100 fractal data matrix.(2)Apply Z-scan to F1 and select any 256 values to get a one-dimensional fractal data vector ZF1 of size 1 × 256. The Z-scan process can be described using a sample matrix of size 4 × 4 as shown in Figure 7.

The other two fractal key streams (F2 and F3) are generated using the same Equation (3) but for two different β values i.e. 0.3 and 0.5. Both the key sequences (F2 and F3) are modified before applying to the row-wise and column-wise pixel substitution process, respectively. The obtained complex number sequences F2 and F3 are converted into double datatype key sequences, i.e. ModF2 and ModF3 according to the given mathematical function as:(6)$$temp\left(i\right)=mod\left(\left(\left(abs\left(F\left(i\right)\right)\right)-floor\left(abs\left(F\left(i\right)\right)\right)\right)\times 10^{8},256\right)$$
(7)ModF(i)=mod(abs(F(i))+temp(i),256)
where i∈[1,2]. A control parameter β provides the flexibility to generate different fractal data sequences. Even a researcher has the option to analyze the randomness of composite fractal function by considering other power values of the Mandelbrot set [53,54]. All generated fractal key sequences exhibit the capability to be used in the cryptosystem design because of its rough structure and irregular scalability in addition to the chaotic properties.

### 3.2. Permutation Phase

An image pixel permutation method deals with the shuffling of pixels within the image. The objective to execute this process is to ensure the reduced adjacent pixel correlation in all directions. The proposed image encryption scheme applied a fast pixel shuffling method to change the position of row and column pixels simultaneously.

#### 3.2.1. Preliminary Steps

Before performing the shuffling process, (1) A Henon map-based pixel scrambling of a plain image P is performed to have more random output upon pixel permutation process execution. The steps of an image pixel scrambling start by executing the Henon map function for M×N times and obtained x and y sequences. Next, apply sorting to the obtained sequences separately and then merge their corresponding sorting indexes. At last, swap the plain image pixels with the merged sorting index sequences and get the scrambled plain image P1.
(8)$$\begin{array}{c} x_{n+1}=1-\left(a\times x_{n}^{2}\right)+y_{n}\\ y_{n+1}=b\times x_{n} \end{array}$$
(9)$$\begin{array}{c} \left[x\_index\right]=sort\left(x\right)\\ \left[y\_index\right]=sort\left(y\right) \end{array}$$
(10)[index]=mod(x_index+y_index,255)+1
(11)P1=swap(P,index)

Further, (2) To permute image pixel, produce a two-dimensional random fractal matrix S using Z-scanned fractal data array ZF1 and a random array R of the same size. Let us suppose a plain image (M×N) needs to be encrypted, then the size of each random array R and ZF1 should be 1×M. Start the process by sorting the array R and ZF1 and stores their index values in R_index and ZF1_index respectively. Now, identify the index value of S using the sorted index of ZF1_index. According to the obtained index value, place the data value of S using the sorted index of random array i.e. R_index. A procedural description is given below:(12)R=randi(1, M)
(13)[R_index]=sort(R)
(14)[ZF1_index]=sort(ZF1)
(15)S_index=mod(n+ZF1_index(m)−1,N)+1
(16)S(m,n)=R_index(S_index)
where m={1,2,…,M} and n={1,2,…,N}. For example, assume ZF1 = {5, 2, 9, 7} and R = {4, 1, 3, 8}. After applying to sort, the obtained ZF1_index and R_index will be {2, 1, 4, 3} and {2, 3, 1, 4} respectively. Now set the value of m to 1 and calculate S_index for all columns using Equation (15). Assign the R_index = {2, 3, 1, 4} value to S(1,1), S(1,2), S(1,3), S(1,4) according to obtained S_index = {3, 4, 1, 2} at the first iteration. As a result, the output of S will be {1, 4, 2, 3}. Repeat the process by executing Equations (15) and (16) for M number of rows and get a two-dimensional random fractal matrix S to do the pixel scrambling in the next step.

#### 3.2.2. Image Permutation Using Random Fractal Matrix

A fast pixel permutation method is proposed to change the pixel position in the row and column simultaneously and reduced the adjacent pixel correlation close to zero. The pixel position shuffling of the previously scrambled plain image P1 takes place with the help of a random fractal matrix S. Identify the row and column index value (x,y) of newly constructed scrambled matrix SP so that the pixel values of P1 can be shuffled accordingly. The detailed steps are discussed as follows:(17)x=mod(S(m,n)−(m−1),M)+1
(18)y=S(x,:)=S(m,n)

The function is repeated for M×N times to get all updated indexes of SP. Now shuffle the elements of P1 according to the modified index values.
(19)SP(x,y)=P1(m,n)

Figure 8 gives a numerical description of how actual plain image pixels will be distributed to its scrambled result. Row-1 of Figure 8a shows the actual index value which will be changed to the index values given in row-2 of the Figure 8a only. In the given example of the 4 × 4 matrix, it can be seen that no row and column number of an index matches with its equivalent modified index, except one index in each matrix row. Figure 8b displays the one-to-one position mapping between the P1 and SP. The color-coded cells help to locate the dispersed cell value of P1 in SP after replacement. Finally Figure 8c represents a matrix view of scrambling process using the sample matrices S, and P1. A simultaneous row and column-wise P1 pixel position change occurs using S and produces a completely random and revised shuffled matrix SP.

### 3.3. Diffusion Phase

A cipher image must be sufficiently sensitive to the slight change in the plain image. A strong diffused system is assumed to have more than 50% change impact on cipher image pixels in the case of plain image pixel change. The proposed diffusion strategy includes pixel-by-pixel substitution in two phases, i.e. row-wise and column-wise. 

The process utilizes the two fractal key streams (ModF2 & ModF3) generated in the previous section to perform the non-linear substitution of the scrambled image pixels SP. The image pixel diffusion is achieved by considering the current SP pixel, same indexed ModF2/ModF3 pixel, and previously diffused pixel (either SP or cipher image). Before using the fractal keystream, shift the key by m or n number of bits at each iteration depending upon the row-wise or column-wise pixel substitution, respectively.

Row-wise pixel substitution: Let us assume m = {1, 2, …, M} and n = {1, 2, …, N}, a row-wise pixel substitution starts by initializing m = 1 and varying n until it reaches the end of the columns. For each m value, shift the ModF2 bits m times before utilizing it in the diffusion process. The mathematical description of the proposed diffusion method is defined as:(20)$$C1\left(m,n\right)=\{\begin{array}{c} SP\left(m,n\right)+shiftF2\left(m,n\right)+SP\left(m,N\right)\;mod\;F,\;\;\;if\;n==1\\ SP\left(m,n\right)+shiftF2\left(m,n\right)+C1\left(m,n-1\right)\;mod\;F,\;\;\;if\;n\;\neq 1 \end{array}$$

Column-wise pixel substitution: In the next step, a column-wise pixel substitution takes place by using the result of the previous section and a new fractal keystream i.e. ModF3. As discussed in the previous section, the diffusion process utilizes an updated ModF3, which will be obtained by shifting the bits n times at each iteration. The diffusion function in this step will work as follows:(21)$$C2\left(m,n\right)=\{\begin{array}{c} C1\left(m,n\right)+shiftF3\left(m,n\right)+C1\left(M,n\right)\;mod\;F,\;\;\;if\;m==1\\ C1\left(m,n\right)+shiftF3\left(m,n\right)+C2\left(m-1,n\right)\;mod\;F,\;\;\;if\;m\;\neq 1 \end{array}$$

Here shiftF2 and shiftF3 denotes the updated ModF2 and ModF3 after bit shifting, respectively. F represents the intensity level of an image pixel such as F = 2 in the case of a binary plain image, and F = 256 if the image pixels are represented by eight bits. A cipher image C will be obtained by repeating the pixel scrambling and pixel diffusion process one more time with the generated sequence C2.

### 3.4. Algorithm

So far, the paper has discussed all the methodologies in detail used to design the proposed cryptosystem. A step-by-step description of the conversion of a plain image to the cipher image and vice-versa can be explained as below.

#### 3.4.1. Encryption Process

Input: A plain image P of size M×N and a composite fractal function CFFGenerate a fractal data matrix F1 of size 100 × 100 by executing CFF for β=0.2, assuming M=N= 256.Apply the Z-scan method to F1 and obtain a one-dimensional array ZF1 of size 1 × 256.Generate two more fractal data matrices F2 and F3 of size M×N for two different β values, i.e. 0.3 and 0.5 using the same CFF.Modify F2 and F3 by implementing an arithmetic operation and obtain ModF2 and ModF3.Obtain P1 by applying the Henon map-based pixel scrambling to P.Generate a two-dimensional random matrix S with the help of ZF1 and a random array R of the same size (1 × 256).Perform image permutation operation on P1 using S and obtain SP.Shift the bits of ModF2 on each row change in row-wise pixel substitution and ModF3 on each column change in column-wise pixel substitution.Execute row-wise pixel diffusion using current pixel of SP, shiftF2, and previously diffused image pixel and get C1.Execute column-wise pixel diffusion using current pixel of C1, shiftF3, and previously diffused image pixel and get C2.Repeat steps (7)–(10) using C2 one more time to get the final cipher image C.Output: An encrypted unintelligible image cipher image C.

#### 3.4.2. Decryption Process

Input: A cipher image C and a composite fractal function CFF
Execute steps (1)–(4) to get initial data matrices to be used in the decryption process as it was in the encryption process.The cipher image C will undergo the pixel diffusion process stated in steps (8)–(10).Perform pixel shuffling of an obtained image using SP.Repeat steps (13) & (14) one more time to get the initial scrambled plain image P1.Finally, to produce the original plain image P, apply the inverse of a Henon map-based pixel scramble operation to P1.

Output: An original plain image P.

### 3.5. Discussion

A secure cryptosystem requires a strong and unidentifiable secret key to protect the information from being hacked while in transmission. The proposed cryptosystem utilized the new composite fractal function to generate the key sequence. Here, three important points justify the effective use of a fractal function-based secret key. (1) A fractal function characterizes a large keyspace in-spite of having a low number of initial values. Thus, this reduces the possibility of a Brute-force attack. (2) Three different fractal data sequences have been generated to be used in pixel scrambling and pixel diffusion separately. (3) The proposed CFF combines the two-generic form of well-known Mandelbrot sets so it has the flexibility to use another combination as well to generate a secret key. A standard confusion-diffusion structure has been followed to design the proposed scheme. The discussed cryptosystem leverages the chaotic characteristics of the fractal function and the randomized plain image pixels because of an additional pixel permutation operation before the actual pixel scrambling process. Other advantages that are achieved by implementing the method: (Each statement is verified in the subsequent simulation results and performance analysis section).

All cipher images have a uniform pixel intensity distribution, which proves the applicability of the method to any type of image representation format.The scheme proposes a two-step diffusion process, i.e. row-wise and column-wise, to get a complete randomized outcome.The pixel confusion and diffusion processes have been repeated two times so that a single bit change in a plain image or a cipher image has a great impact on the corresponding encryption or decryption phase, respectively.A random fractal data matrix will always produce a different shuffled image, thus it can resist the well-known attacks such as chosen-plaintext, differential, and statistical attacks.All three secret key matrices are derived using the discussed CFF so any slight change in the function will change the key matrices completely, and correspondingly, the cipher image.

Till now, we have seen the positive side of the given scheme, but it also portrays a drawback during image recovery from noise and data loss attack. This is due to the impact of repetitive execution of fractal function-based image permutation and diffusion. Because of the high sensitivity towards its initial value, a fractal function-based key will result in an entirely changed decrypted image on any small change in the cipher image (such as noise attack or data loss). Therefore, a legitimate receiver can face difficulty to recover the original image from a noisy or lost data image. In the future, a robust cryptosystem based on a fractal function with multiple rounds can be discussed, which would be secure from the noise and data loss of course.

## 4. Simulation Results and Performance Analysis

The section discusses the simulation results of the proposed image encryption, executed using MATLAB (2016b) software. The purpose of any image encryption algorithm is to come up with the strategy to be able to produce a noisy cipher image that completely differs from its corresponding plain image. It helps to make sure that the hacker would not be succeeded to penetrate the cipher image through the statistical analysis of it. Further, various suggested performance tests have performed to verify the security, suitability, and robustness of the new CFF-based image encryption algorithm [55].

To verify the feasibility of the proposed system, few sample images were chosen to test the performance visually. The method can be applied to any image representation format like 2-bit, 8-bit, 16-bit, and 24-bit. A color image has a 24-bit need to split three channels (R, G, B) of 8-bit each before applying the image encryption algorithm. Each divided channel is a 2D matrix. Execute the given image encryption method to each color component separately and get the required encrypted/decrypted images. At last, merge all the three image channels (R, G, B) to get the desired result at both ends.

Figure 9 shows the gray, binary, and color plain images and their corresponding cipher and decrypted images. As can be seen, the cipher image has a noise-like visual, it is hard to identify the plain image by analyzing it. Similarly, the reconstructed image without any loss of information proved the efficiency of the decryption algorithm as well. The subsequent section will evaluate the security level of the proposed scheme to ensure the system’s resistivity against various ciphertext attacks.

### 4.1. Histogram Analysis & Chi-Square Test

A histogram depicts the image pixel intensity distribution throughout the axes. A plain image with highly correlated pixels allowed the hacker to infer the system. That is the reason: a cipher image must have reduced or almost not correlated image pixels. The condition is achieved by distributing the image pixel frequency uniformly. The histogram of a plain image and cipher image is shown in Figure 10. The even distribution of image pixels makes it difficult for the hacker to steal valid statistical information from the cipher image. Hence, it proved the effectiveness of the system against the statistical attack.

Besides the visual competency of the system, a quantitative analysis of the randomness is done by implementing the chi-square test. The mathematical function to compute the chi-square value can be given as:(22)$$\chi^{2}=\sum^{\;}\frac{{\left(O_{i}-E_{i}\right)}^{2}}{E_{i}}$$
where $(O_{i})\;$and ($E_{i})$ are observed and expected pixel intensity frequency respectively. According to [56], a random gray image should have a chi-square value less than the standard value 293.2478 at significance level 0.05. Table 1 lists all chi-square values computed for the considered sample gray images and their respective cipher images. The outcome indicates a better uniformity of pixel intensity, a requirement to design a secure cryptosystem.

### 4.2. Key Space

A cryptosystem must have a sufficiently large keyspace to resist the brute-force attack. According to [57], a keyspace equal to or larger than $2^{100}$ is good enough to make brute-force attack infeasible. A plain image pixel permutation operation is executed using the Henon map, which requires two variables a and b. Additionally, the secret key sequences F2 and F3 used in the diffusion process are derived by executing a new CFF, which requires six parameters, i.e. x, y, initial value z, number of iterations n, and β (0.3 and 0.5). Similarly, a different key sequence is generated by considering different n and β (0.2) to shuffle the image pixels. So, a total of 10 double-precision values will be used in the key sequence generation process. Considering the fact given in IEEE floating-point standard, the computational precision of a double datatype variable is $10^{-15}$. Accordingly, the total available keyspace will be $\left(10^{15*10\;}\right)\approx 2^{495\;}$. The obtained keyspace value is quite greater than the required $2^{100}$ to secure the proposed cryptosystem against the brute-force attack.

### 4.3. Adjacent Pixel Correlation Analysis

A correlation coefficient depicts the relationship strength between the adjacent image pixels. This test is used to evaluate the encryption quality by relating the pixels of plain and cipher images horizontally, vertically, and diagonally. An ideal encryption algorithm should be able to produce an encrypted image with no correlation between the adjacent pixels in all directions. In such a scenario, an intruder would not get sufficient information about the image structure to recover the original image. The pixels correlation coefficient of plain image and cipher image can be calculated as:(23)$$M\left(x\right)=\frac{1}{N}\overset{N}{\underset{i=1}{\sum }}x_{i}$$
(24)$$var\left(x\right)=\frac{1}{N}\overset{N}{\underset{i=1}{\sum }}{\left(x_{i}-M\left(x\right)\right)}^{2}$$
(25)$$cov\left(x,y\right)=\frac{1}{N}\overset{N}{\underset{i=1}{\sum }}\left(x_{i}-M\left(x\right)\right)\left(y_{i}-M\left(y\right)\right)$$
(26)$$cc=\frac{cov\left(x,y\right)}{\sqrt{var\left(x\right)}*\sqrt{var\left(y\right)}}$$
where function M(x) is the mean, var(x) is the variance, and cov(x,y) is the covariance of selected N pixels. Finally, function cc estimates the correlation coefficient between adjacent pixels x and y of the plain and cipher image. A cc value of nearly one indicates an excellent similarity, while the value of zero explicates no correlation between the adjacent image pixels. Therefore, the cc ≈ 0 conferred that the plain and cipher image contents are highly dissimilar to each other.

The given approach is evaluated for different gray images of size 256 × 256. The association among neighboring pixels can be measured by plotting pixels distribution of plain and cipher images in all directions, which are shown in Figure 11. The distribution pattern concludes whether the pixels are correlated or not. A diagonal line pixel pattern depicts the excessive amount of adjacent pixels similarity. On the other side, and evenly distributed, cipher image pixel signifies the reduced pixel correlation in all directions. The quantifiable investigation of the correlation coefficient of each image is shown in Table 2 in all possible directions. The plain image and a cipher image have the anticipated cc values in horizontal, vertical, and diagonal directions. The test authorized that the system meets the requirement of highly non-correlated images before and after encryption.

### 4.4. Correlation Between Plain and Cipher Image

The Pearson correlation coefficient (PCC) quantitively describes the linear relationship between two images. The PCC value is computed by using the given function as:(27)$$PCC\left(x,y\right)=\frac{N\sum_{i=1}^{N}(x_{i}y_{i})-\sum_{i=1}^{N}(x_{i})\sum_{i=1}^{N}(y_{i})}{\sqrt{\left(N\sum_{i=1}^{N}{\left(x_{i}\right)}^{2}-(\sum_{i=1}^{n}{\left(x_{i}\right)}^{2}\right)\;\left(N\sum_{i=1}^{N}{\left(y_{i}\right)}^{2}-(\sum_{i=1}^{n}{\left(y_{i}\right)}^{2}\right)}}$$
where x and y are two variables that define two neighboring pixels of an image, and N refers to the number of pixel pairs. Here, x and y are assumed to be the plain image and cipher image pixel respectively. The PCC value varies between [−1, 1]. Here, −1 indicates an inverse relationship between the variables, whereas +1 shows a direct relationship and highly correlated variables. The value close to zero indicates that the plain and cipher image are not correlated to each other, which is an expected outcome of a secure cryptosystem. Table 3 lists the correlation coefficient value of tested sample images.

### 4.5. NPCR and UACI Tests

To test the robustness of the proposed system against the differential attack, the measurement of the impact of the minor alteration in the plain image of its corresponding cipher image is carried out [58]. The number of pixels changing rate (NPCR) test is implemented to check the influence capacity of plain image pixel change. An NPCR value illustrates the pixel difference between the two encrypted images at the same position, whose plain image is differed by only one bit. To evaluate the probable impact, consider two encrypted images ($C_{1}$ and $C_{2}$) obtained before and after the one-pixel change in plain image. The NPCR can be calculated using the given formula as:(28)$$NPCR=\frac{1}{M*N}\overset{M}{\underset{i=1}{\sum }}\overset{N}{\underset{j=1}{\sum }}D\left(i,\;j\right)\times 100\%$$
where D(i, j) is calculated as:(29)$$D\left(i,j\right)=\{\begin{array}{c} 0,\;\;\;\;\;\;\;\;\;if\;C_{1}\left(i,j\right)=C_{2}\left(i,j\right)\\ 1,\;\;\;\;\;\;\;\;\;if\;C_{1}\left(i,j\right)\neq C_{2}\left(i,j\right) \end{array}$$

UACI or Unified average changed intensity test calculates the pixel intensity difference between two images. Here the test is applied to two encrypted images $C_{1}$ and $C_{2}$ having size 256 × 256. The mathematical formula to calculate UACI value as follows:(30)$$UACI=\frac{1}{M*N}\overset{M}{\underset{i=1}{\sum }}\overset{N}{\underset{j=1}{\sum }}\frac{C_{1}\left(i,j\right)-C_{2}\left(i,j\right)}{255}\times 100\%$$

In the proposed method, a pixel of the plain image $P_{1}$ is randomly selected and changed to one to obtain a new plain image $P_{2}$. Now the NPCR and UACI tests are implemented on the corresponding encrypted images $C_{1}$ and $C_{2}$. The evaluation of the projected technique is done by calculating the NPCR and UACI values for all test images and listed in Table 4. The accepted critical values of NPCR and UACI tests are NPCR > 99.5693% and UACI ∈ [33.22–33.70] respectively. The outcome demonstrates that both test values are approaching the ideal values due to the 1-bit alteration in the plain image.

### 4.6. Information Entropy

The term “entropy” was originally used by a scientist named Claude Shannon in 1949 [12] to find out the degree of chaos. The tool is widely used to test the gray value distribution probability. Here the global Shannon entropy (GSE) and local Shannon entropy (LSE) of an image are calculated to measure the unpredictability and randomness of the suggested encryption system. The formula to calculate the entropy of an image is defined as:(31)$$H\left(s\right)=-\overset{2^{n}-1}{\underset{i=0}{\sum }}p\left(x_{i}\right)log_{2}\left[p\left(x_{i}\right)\right]$$
where $P\left(x_{i}\right)$ is the discrete probability density function for the pixel value ranges (0–255). Both entropies will be calculated using the same function except that the GSE is calculated for the entire image, whereas LSE considers randomly selected non-overlapping blocks of an image [59]. An image having 256 possible values will show the maximum entropy equal to eight (i.e. $2^{8}=256)$. To calculate LSE, consider k non-overlapping image blocks and calculate the Shannon entropy of each block using Equation (31). Now compute the mean entropy value of all k image blocks to get LSE. An accepted value of LSE falls between the range [7.901516–7.903423] to pass the randomness test. Table 5 lists the global Shannon entropy and local Shannon entropy of the encrypted images along with plain image entropy. It is shown that the calculated GSE and LSE values of cipher images are close to the expected values. Therefore, the proposed scheme has successfully passed the randomness test for a variety of different test images.

### 4.7. MSE, MAE, PSNR and SSIM Tests

The parameters used to check the encryption quality of an image encryption system using its pixel’s disparity are mean absolute error (MAE), mean square error (MSE) and peak signal to noise ratio (PSNR). The MAE value is calculated to check the average absolute pixel difference between a plain image P and a cipher image C of size M×N. The MAE function can be defined as:(32)$$MAE=\frac{1}{M*N}\overset{M}{\underset{i=1}{\sum }}\overset{N}{\underset{j=1}{\sum }}\left|C\left(i,j\right)-P\left(i,j\right)\right|$$

The MSE value refers to the cumulative square error measure, which is calculated by measuring the squared difference between a plain image and a cipher image. Moreover, the PSNR value analyzes the image fidelity by calculating the ratio between the maximum pixel intensity and the MSE value. The MSE and PSNR values will be calculated using the following functions:(33)$$MSE=\frac{1}{M\times N}\overset{M}{\underset{i=1}{\sum }}\overset{N}{\underset{j=1}{\sum }}{\left(P\left(i,j\right)-C\left(i,j\right)\right)}^{2}$$
(34)$$PSNR=20log_{10}\left(\frac{{\left(2^{L}-1\right)}^{2}}{\sqrt{MSE}}\right)db$$
where L represents the bit-depth of the used image. Another structural similarity index measurement (SSIM) test quantifies the perceptual difference between two images. The function to calculate SSIM is defined as:(35)$$SSIM=\frac{\left(2\bar{P}\bar{C}+S_{1}\right)\;\left(2\sigma_{PC}+S_{2}\right)}{\left({\bar{P}}^{2}+{\bar{C}}^{2}+S_{1}\right)\;\left(\sigma_{P}^{2}+\sigma_{C}^{2}+S_{2}\right)}$$
where
(36)$$\bar{P}=\frac{1}{M\times N}\overset{M}{\underset{i=1}{\sum }}\overset{N}{\underset{j=1}{\sum }}P\left(i,j\right)$$
(37)$$\bar{C}=\frac{1}{M\times N}\overset{M}{\underset{i=1}{\sum }}\overset{N}{\underset{j=1}{\sum }}C\left(i,j\right)$$
and
(38)$$\sigma_{P}=\frac{1}{M\times N}\overset{M}{\underset{i=1}{\sum }}\overset{N}{\underset{j=1}{\sum }}{\left[P\left(i,j\right)-\bar{P}\right]}^{2}$$
(39)$$\sigma_{C}=\frac{1}{M\times N}\overset{M}{\underset{i=1}{\sum }}\overset{N}{\underset{j=1}{\sum }}{\left[C\left(i,j\right)-\bar{C}\right]}^{2}$$
(40)$$\sigma_{PC}=\frac{1}{M\times N}\overset{M}{\underset{i=1}{\sum }}\overset{N}{\underset{j=1}{\sum }}\left[P\left(i,j\right)-\bar{P}\right]\;\left[C\left(i,j\right)-\bar{C}\right]$$
where $\bar{P}$ and $\bar{C}$ are mean values of plain and cipher image respectively, $\sigma_{P}$ and $\sigma_{C}$ are the standard deviation of plain and cipher image respectively, and $\sigma_{PC}$ is the cross-correlation of the plain and cipher image.$\;S_{1}$ and $S_{2}$ are used for stability and can be calculated as $S_{1}={\left(0.01\times L\right)}^{2}$ and $S_{2}={\left(0.03\times L\right)}^{2}$. Here, L represents the image intensity value, which is 255 for a gray image. 

A PSNR value equal to infinity indicates that both considered images are the same and an SSIM value close to zero represents two structurally different images. Whereas a higher MAE and MSE value ensure the robustness of the proposed system against the cyber-attack. This shows that to achieve a secure cryptosystem, the PSNR and SSIM values should be low, and the MAE and MSE values should be on the higher side. To prove the validity of the given scheme, all four tests have been conducted on a few test images and the results are listed in Table 6. The result states that the encryption quality of all images follows the standards of being a secure cryptosystem.

### 4.8. Performance Analysis

The proposed method is implemented on a computer with the specifications: MATLAB R2016b platform and system configuration Intel® Atom™ x7-z8700 CPU @1.60GHz and 4 GB RAM. Various parameters such as encryption time (ET), encryption throughput (ETP), and number of cycles per bytes (NCB) are used to measure the efficiency of the system.

The given method employed two rounds of confusion-diffusion to achieve a secure and reliable cryptosystem. It is obvious to have longer execution time proportional to the number of rounds to be taken to execute the system. A trade-off between time and security indicates the acceptance of long-running time considering the excellent security requirement. Further, the ETP is computed by estimating the number of bytes executed per second. The mathematical function for ETP and NCB can be defined as:(41)$$ETP=\frac{Image\;size\;\left(bytes\right)}{Encryption\;time\;\left(sec\right)}$$
(42)$$NCB=\frac{CPU\;speed\;\left(Hz\right)}{Encryption\;throughput\;\left(bytes\right)}$$

The performance analysis of the anticipated system in terms of ET, ETP, and NCB has been done for the sample images and the results are listed in Table 7. The efficiency of the given scheme is comparable with the other fractal-based cryptosystem, where a secure system is more preferable than the time constraint environment. 

### 4.9. Noise and Data Loss Attack Analysis

An information transmission channel is always prone to noise and data loss attacks. It is required that an efficient encryption algorithm still manages to recover the original image without losing much information. However, it is impossible to decrypt an errored image into its original form even with the accurate secret key. Hence, a marginal amount of blurred decrypted image is also acceptable.

The proposed scheme executes two rounds of image pixel confusion and diffusion asymmetrically. The diffusion process also runs in a row-wise and column-wise manner. An asymmetric way of pixel substitution includes the encryption of one plain image byte using the current plain image byte along with the previous ciphered image byte. Whereas in decryption, a cipher image byte will be processed with its previous ciphered byte but no plain image data. Also, for each row/column, bit shifting of random fractal data matrix occurred to make the process more complex. The complete diffusion process illustrates that the impact of one-bit change in the plain image will influence all pixels in the corresponding cipher image. Further, this impact will be followed in the subsequent second round of execution. Though the given method succeeded to recover a high-quality image against the data loss attack, it was not robust enough against the noise attack.

Figure 12 shows the actual cipher image, noisy cipher image with 2% of salt & pepper noise, and three cipher images with different percentages of data loss along with their corresponding decrypted images, respectively. It can be observed that the recovered images lost some information but are still visualized by the human eye.

### 4.10. Key Sensitivity Analysis

In this paper, the key sensitivity is quantitatively analyzed using the number of bits change rate (NBCR), the number of pixels changing rate (NPCR), and unified average changed intensity (UACI) tests. The NBCR of two images A and B are computed as:(43)$$NBCR=\frac{Ham\_dis\left(A,B\right)}{bit\_len}$$
where Ham_dis calculates the Hamming distance between A and B, and bit_len is the bit length of an image (either A or B). An NBCR value close to 50% verifies the effectiveness of the key sensitivity impact on the given scheme.

To test the key sensitiveness, select a pixel of ModF2 and ModF3 randomly and change it to 1. Now encrypt the plain image:
using a correct ModF2 and ModF3 and get C1using a bit changed ModF2 and a correct ModF3 and get C2using a correct ModF2 and a bit changed ModF3 and get C3

Similarly, decrypt the correct cipher image using the same key combinations discussed above and subsequently get three decrypted images D1, D2, and D3. The next step is to calculate the NBCR value of (C1,C3) and (C2,C3) to measure the key sensitivity impact on an encryption method. Repeat the same step on decrypted images and get the NBCR value of (D1, D3) and (D2,D3). The computed NBCR value in both processes is listed in Table 8.

Two tests, NPCR and UACI tests, are also implemented to observe the key sensitivity behavior in both encryption and decryption processes. The quantitative results of NPCR and UACI tests are given in Table 9. Figure 13 also represents the key sensitivity exploration results in the form of an encrypted/decrypted image, their respective histogram, and the image difference between two encrypted/decrypted images. It is indicated that a single pixel change in either secret key can produce a completely unpredictable outcome. This shows the excellent key sensitivity of the proposed scheme.

### 4.11. Chosen-Plaintext Attack Analysis

A chosen-plaintext attack is a kind of threat to the system in which an attacker randomly selects a plaintext and encrypts it using an arbitrary secret key. By analyzing the system behavior and corresponding ciphertext, the attacker can infer some secret information to obtain actual plaintext. A secure cryptosystem must be able to fail the unauthorized access of the information by resisting the chosen-plaintext attack. To achieve it, an encryption algorithm must change the plain image data in multiple rounds to get a completely randomized output. The proposed method modifies the plain image by shuffling the pixel positions using the Henon map one time. Again, two rounds of pixel confusion and diffusion process have been implemented to the shuffled image. As a result, the encryption algorithm produces different cipher images on encrypting an identical plain image multiple times using the same secret key. To demonstrate the applicability of the concept, the number of bit change rate (NBCR) of two cipher mages is calculated, which are obtained by encrypting the same image twice using the same secret key. The test result is listed in Table 10. The NBCR value close to 50% indicates that both cipher images are completely different from each other. Hence, the given system can defend the chosen-plaintext attack successfully.

### 4.12. Image Autocorrelation Analysis

An image correlation test is executed by comparing all possible pixel pairs to compute the similarity index of both pixels as a function of distance and direction of separation [60]. An efficient way to measure the image autocorrelation is via a fast Fourier transform using the Weiner–Khinchin theorem. The image autocorrelation function (ACF) can be defined as:(44)$$F^{-1}\left[ACF\left(a,b\right)\right]=S\left(i\right)={\left|F\left[i\left(x,y\right)\right]\right|}^{2}$$
where i(x,y) represents the image intensity at position (x,y), and a and b show the distance from the corresponding x and y position. S(i) is a power spectrum and calculated by squaring the magnitude of the Fourier transform F of an image i. According to the theorem, the Fourier transform of the autocorrelation of an image is equal to the inverse Fourier transform of S(i). Therefore, the autocorrelation can also be calculated as:(45)$$ACF\left(a,b\right)=F^{-1}\left\{F\left[i\left(M,N\right)\right]\times \bar{F}\left[i\left(M,N\right)\right]\right\}$$
where $\;\bar{F}$ is the conjugated Fourier transform of an image of size M×N. The test is implemented to the original image as well as to the cipher image. A highly correlated original image autocorrelation is expected to be wavy and spikes at the center of its graphical representation. On the contrary, cipher image pixels are expected to be minimally correlated with each other, so the graph must have a flat and uniform image autocorrelation structure. The results of the original image and cipher image autocorrelation are shown in Figure 14; they verify the expected outcome of an efficient cryptosystem.

### 4.13. Decryption Error

The probability of decryption error is mainly influenced by the randomness introduced to generate a secret key or in the encryption method. A reliable cryptosystem ensures that the received information will be identical to the sending information irrespective of any kind of random data insertion. This feature is more crucial in some applications like medical, finance, military, geographical, astrological, and many more. For a plain image P and corresponding cipher image C of size M×N, the decryption error DE can be computed using the given function as:(46)$$DE=\frac{100}{M\times N}\overset{M}{\underset{i=1}{\sum }}\overset{N}{\underset{j=1}{\sum }}Q\left(i,j\right)$$
where
(47)$$Q\left(i,j\right)=\{\begin{array}{c} 0,\;if\;P\left(i,j\right)=C\left(i,j\right)\\ 1,\;if\;P\left(i,j\right)\neq C\left(i,j\right) \end{array}$$

In the case of a fully recovered decrypted image, the decryption error must be equal to zero percentage. It indicates that the decrypted image is identical to the original plain image. The decryption error test is successfully executed to the various test images.

### 4.14. Performance Comparison with Existing Work

To assess the efficiency of the projected encryption method, its performance is compared with several, mostly fractal-based image encryption methods. A comparative analysis with existing methods is done by considering the same image, i.e. Lena (256 × 256). The comparison is done by considering keyspace, few differential attack measures, global Shannon entropy, and adjacent correlation coefficient.

As shown in Table 11, the discussed method has the flexibility to select a key from the available keyspace that is sufficiently large to resist the brute-force attack. The NPCR and UACI values of the proposed method are 99.63 and 33.54, respectively, which is the largest among all references. It also proved the system resistance against the differential attack. An ideal entropy value is expected to be much closer to eight. The value reported in Table 11 indicates that the obtained entropy value passed the pixel randomness test. The correlation coefficient value is also compared with the existing approaches in all three directions. Presented data reveal that the encrypted pixel values are approaching zero. It signifies the desired result of reduced or no correlation between the adjacent pixels. Therefore, all the discussed scenarios verify the suitability of the proposed scheme in the secure digital image transmission over the unsecured network.

## 5. Conclusions and Future Work

This paper proposed a composite fractal function (CFF) consisting of two different Mandelbrot set functions as a seed map and a control parameter to enhance the non-linear dynamics. Many performance tests such as self-similar image structure, fractal dimension, chaotic trajectory, and Lyapunov exponent were implemented to evaluate the significance of the new fractal function over the original seed function. Using the newly constructed fractal, a secure, simple yet complex cryptosystem was designed. The suggested algorithm displayed increased randomness by applying a Henon map-based plain image pixel scrambling and a random fractal matrix generation using a z-scanned fractal sequence. Additionally, a fractal key sequence was employed to both pixel-based permutation and diffusion phase, but in three different organizations. The simulation result showed the suitability of the given scheme to the different digital image formats such as color, gray, and binary image. The fractal function performance analysis demonstrated that the suggested CFF possesses all desired chaotic properties such as randomness, unpredictability, initial value sensitivity, ergodicity, large chaotic trajectory, and a complex self-similar structure. Moreover, the experimental results of the image cipher also indicated a significantly superior performance in terms of the large keyspace, high key sensitivity, expected global Shannon entropy, local Shannon entropy, adjacent pixel correlation, expected image autocorrelation, NPCR and UACI value, and zero decryption error. However, the method is susceptible to noise and data loss attacks due to the two rounds of pixel confusion-diffusion execution, but it can be considered to some extent to achieve an extremely sensitive cryptosystem. In the future, the robustness of the system against the noise and data loss attacks will be addressed.

## Figures and Tables

**Figure 1 jimaging-06-00070-f001:**
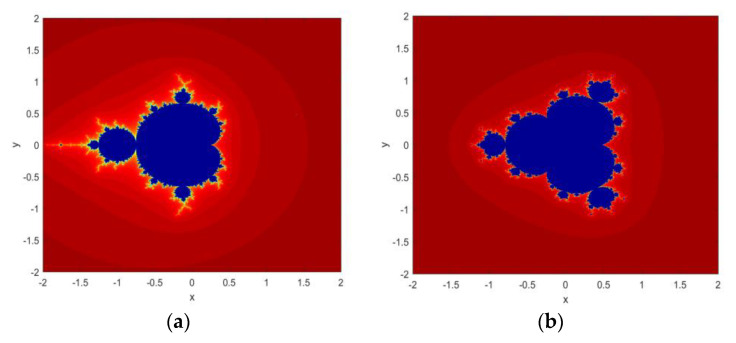
Fractal images generated from MS: (**a**) Equation (1); (**b**) Equation (2).

**Figure 2 jimaging-06-00070-f002:**
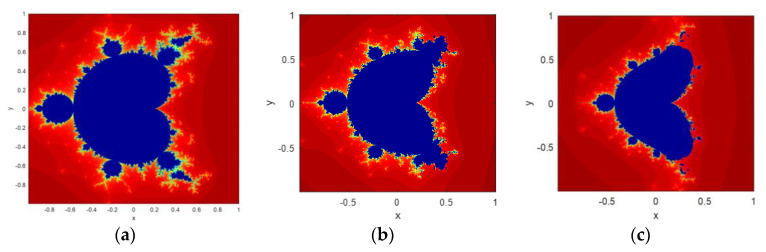
Fractal images generated from CFF: (**a**) β = 0.2; (**b**) β = 0.3; (**c**) β = 0.5.

**Figure 3 jimaging-06-00070-f003:**
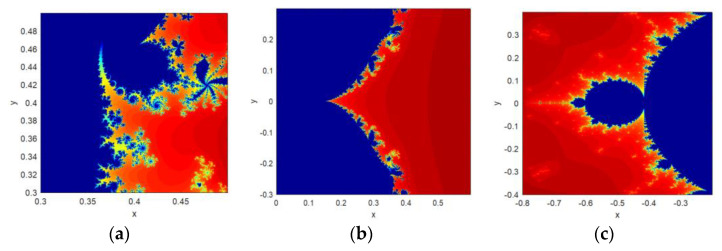
Zoomed version of fractal images generated from CFF: (**a**) β = 0.3; (**b**) β = 0.5; (**c**) β = 0.5.

**Figure 4 jimaging-06-00070-f004:**
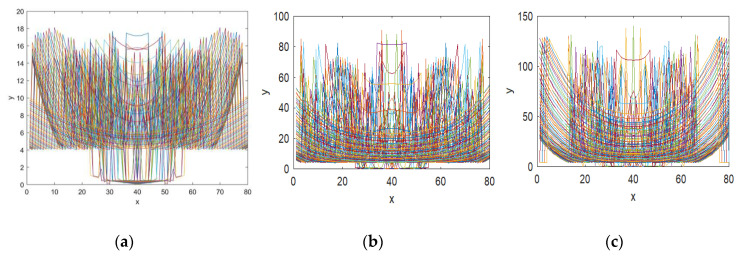
Fractal image trajectory map generated from: (**a**) MS function; (**b**) CFF with β = 0.3; (**c**) CFF with β = 0.5.

**Figure 5 jimaging-06-00070-f005:**
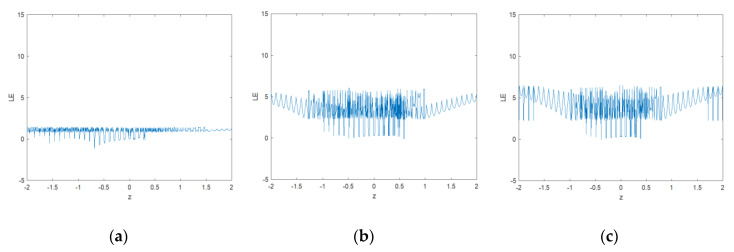
Lyapunov map of fractal images generated from: (**a**) MS function; (**b**) CFF with β = 0.3; (**c**) CFF with β = 0.5.

**Figure 6 jimaging-06-00070-f006:**
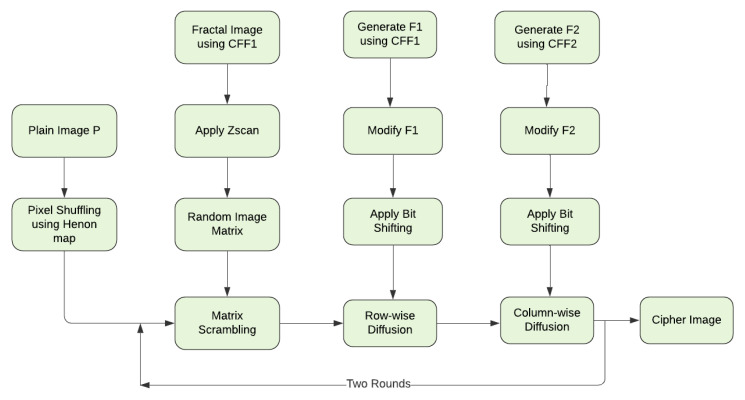
Proposed Image encryption method block diagram.

**Figure 7 jimaging-06-00070-f007:**
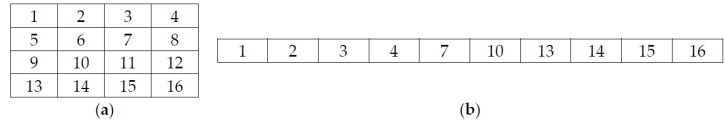
(**a**) Sample CFF matrix; (**b**) Z-Scanned one-dimensional array.

**Figure 8 jimaging-06-00070-f008:**
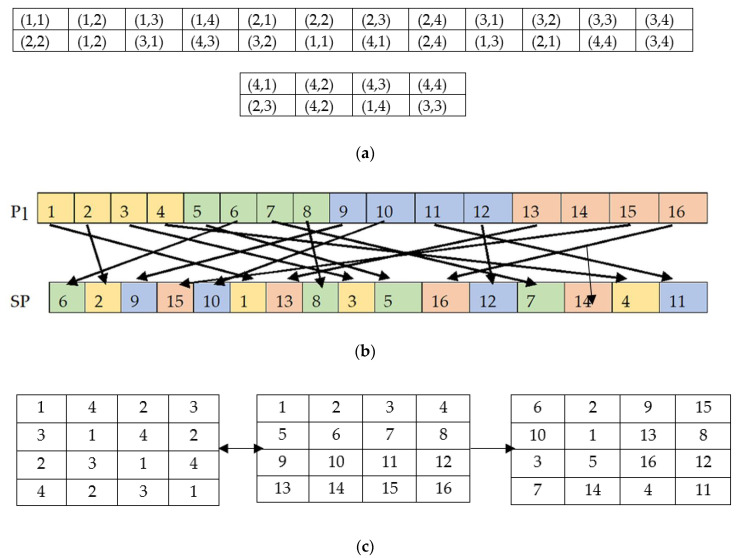
Scrambling process example using 4 × 4 matrix. (**a**) Actual pixel position and its corresponding shuffled pixel position; (**b**) pixels mapping between P1 and SP; (**c**) Matrix view of the scrambling process using S, P1 and SP respectively.

**Figure 9 jimaging-06-00070-f009:**
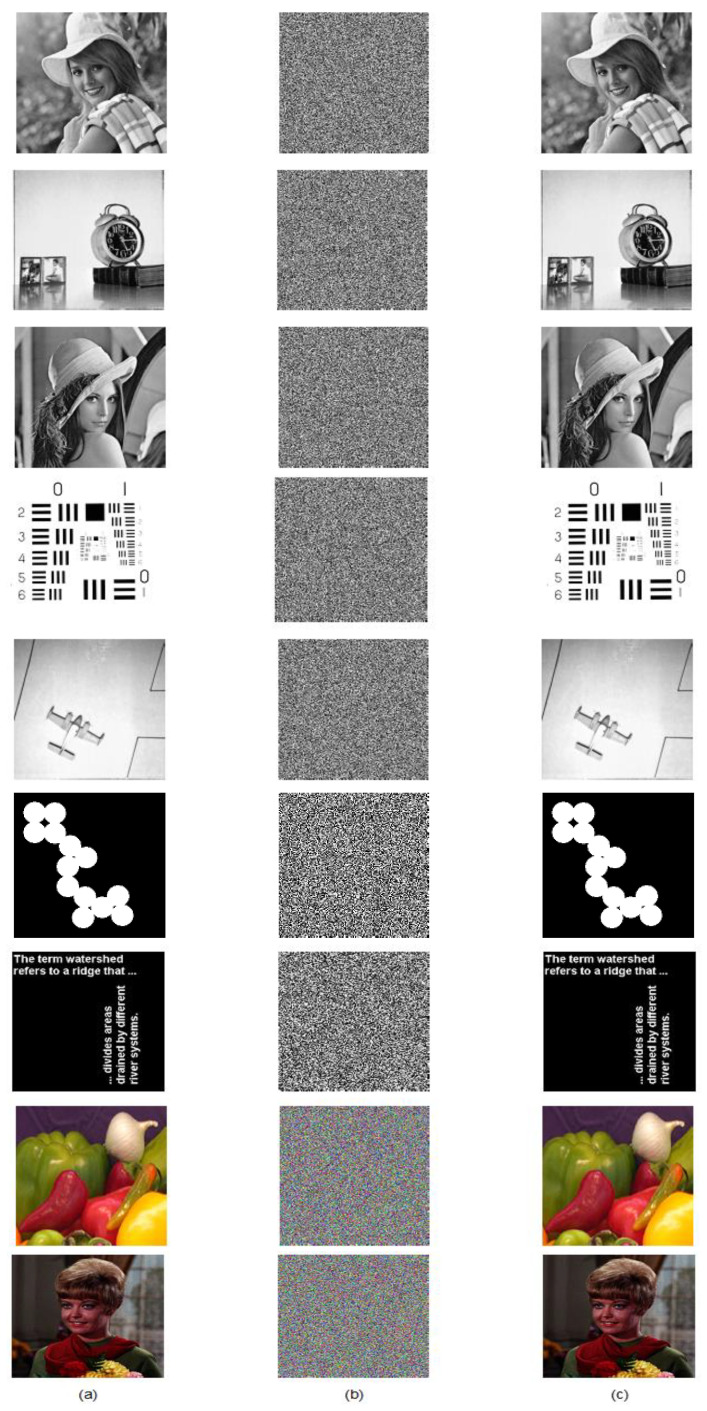
(**a**) Plain image; (**b**) Encrypted image; (**c**) Decrypted image.

**Figure 10 jimaging-06-00070-f010:**
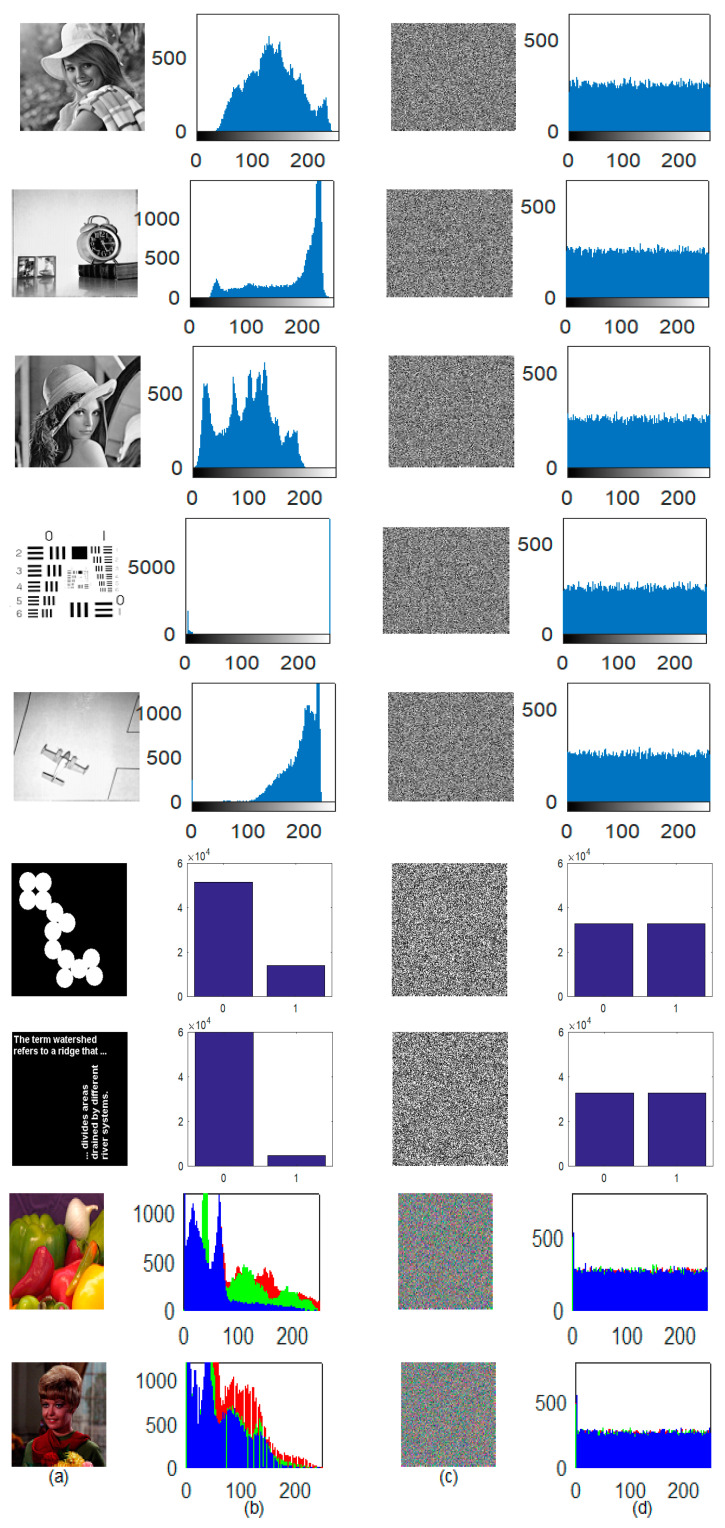
(**a**) Plain image; (**b**) Histogram of (**a**); (**c**) Cipher image; (**d**) Histogram of (**c**).

**Figure 11 jimaging-06-00070-f011:**
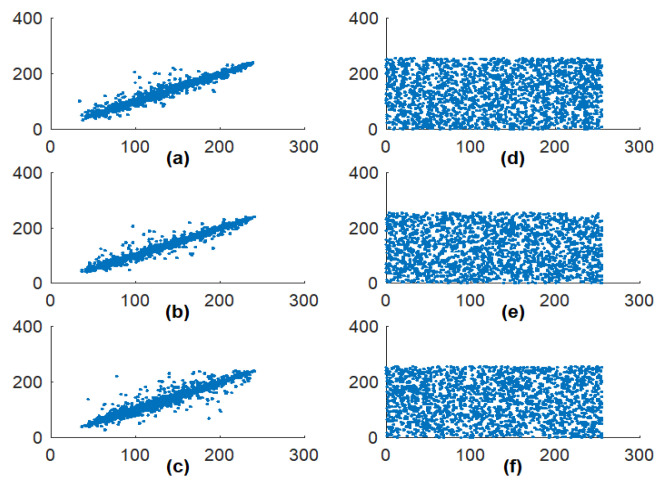
Elaine image pixel distribution in a different direction. (**a**–**c**) Original image pixel distribution in horizontal, vertical, and diagonal direction; (**d**–**f**) Cipher image pixel distribution in horizontal, vertical, and diagonal direction.

**Figure 12 jimaging-06-00070-f012:**

(**a**) Actual cipher image and decrypted image; (**b**) Noisy cipher image with 2% salt and pepper noise and its decrypted image; (**c**–**e**) Cipher images and their corresponding decrypted images with different percentages of data loss.

**Figure 13 jimaging-06-00070-f013:**
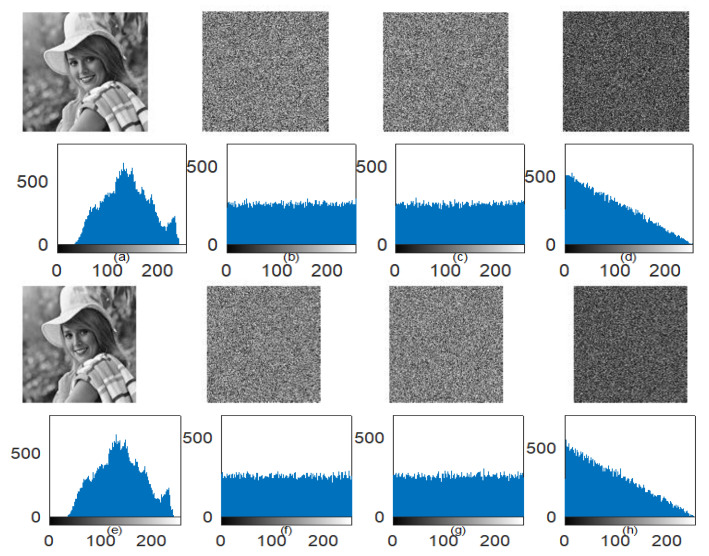
(**a**–**d**) Encryption sensitivity using the image and its histogram, (**a**) Elaine image; (**b**) Encrypted image using correct secret key; (**c**) Encrypted images using modified secret key; (**d**) Image difference between encrypted images using correct and modified secret key (**e**–**h**); Decryption sensitivity using an image and its histogram; (**e**) Decrypted images using the correct secret key; (**f**–**g**) Decrypted images using two different modified secret keys; (**h**) Image difference between decrypted images using two different modified secret keys.

**Figure 14 jimaging-06-00070-f014:**
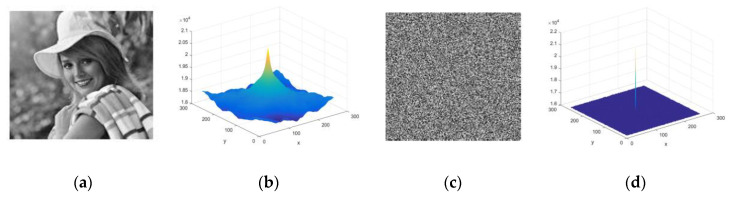
Image autocorrelation plots: (**a**) Original image; (**b**) Autocorrelation of original image; (**c**) Cipher image; (**d**) Autocorrelation of cipher image.

**Table 1 jimaging-06-00070-t001:** Chi-square values of sample images.

Images	Elaine	Lena	Clock	5.1.13	5.1.11
Plain Image	36011.91	42101.414	281805.5625	70357.64	220592.7
Cipher Image	246.9	223.35	241.35	253.8	229.36

**Table 2 jimaging-06-00070-t002:** Pixel correlation coefficient values of sample images.

Images	Plain Image			Cipher Image		
	Horizontal	Vertical	Diagonal	Horizontal	Vertical	Diagonal
Elaine	0.9707	0.9755	0.9509	−0.002	0.0009	−0.0051
Lena	0.9541	0.9778	0.9321	−0.002	0.0017	−0.0027
Clock	0.9564	0.9740	0.9389	−0.0026	0.0021	0.0015
5.1.13	0.8721	0.8667	0.7561	0.0003	−0.0011	−0.0052
5.1.11	0.9570	0.9365	0.8926	−0015	0.0026	−0.0024

**Table 3 jimaging-06-00070-t003:** Correlation Coefficient value between plain and cipher images.

Images	Elaine	Lena	Clock	5.1.13	5.1.11
PCC	−0.0014	0.0013	0.0035	−0.0012	0.0021

**Table 4 jimaging-06-00070-t004:** NPCR and UACI values of cipher image.

Images	NPCR (%)	UACI (%)
Elaine	99.62	33.55
Lena	99.63	33.54
Clock	99.6	33.39
5.1.13	99.62	33.56
5.1.11	99.64	33.47

**Table 5 jimaging-06-00070-t005:** GSE and LSE values of sample cipher images.

Images	Cipher Image	
	GSE	LSE
Elaine	7.9973	7.9036
Lena	7.9975	7.9027
Clock	7.9972	7.9023
5.1.13	7.9972	7.9017
5.1.11	7.9975	7.9014

**Table 6 jimaging-06-00070-t006:** MAE, MSE, and PSNR values of sample cipher images.

Image	MAE	MSE	PSNR	SSIM
Elaine	72.49	7641.788	9.2992	0.0091
Lena	76.34	8613.222	8.7766	0.0093
Clock	89.98	12146.04	7.2866	0.0102
5.1.13	124.23	20924.15	4.9271	0.0082
5.1.11	85.31	10899.16	7.7516	0.0096

**Table 7 jimaging-06-00070-t007:** ET, ETP, and NCB of sample images.

Images	Elaine	Lena	Clock	5.1.13	5.1.11
ET (sec)	1.31	1.34	1.28	1.37	1.45
ETP (Mbps)	0.0477	0.0466	0.0488	0.0456	0.0431
NCB (Hz/bytes)	31982	32714.85	31250	33447.27	35400.39

**Table 8 jimaging-06-00070-t008:** NBCR values of sample images in the encryption and decryption processes.

Images	Encryption Process		Decryption Process	
	NBCR (C1, C3)	NBCR (C2, C3)	NBCR (D1, D3)	NBCR (D2, D3)
Elaine	49.99	49.98	50.05	50
Lena	49.97	50	49.99	50.01
Clock	50.09	49.94	50.08	49.94
5.1.13	49.93	50	50	49.89
5.1.11	49.99	50.02	49.98	49.97

**Table 9 jimaging-06-00070-t009:** NPCR and UACI values in the key sensitivity analysis.

Images	NPCR1 (%)	UACI (%)	NPCR2 (%)	UACI (%)
Elaine	99.61	33.37	99.64	33.49
Lena	99.63	33.49	99.61	33.5
Clock	99.62	33.46	99.61	33.48
5.1.13	99.60	33.43	99.61	33.48
5.1.11	99.60	33.39	99.62	33.46

**Table 10 jimaging-06-00070-t010:** Chosen-plaintext attack analysis result.

Images	Elaine	Lena	Clock	5.1.13	5.1.11
NBCR	50.01	50.05	50.02	50.20	49.97

**Table 11 jimaging-06-00070-t011:** Comparison of keyspace, NPCR, UACI, GSE, and correlation coefficient values between the proposed work and the existing methods.

Algorithm	Key Space	NPCR (%)	UACI (%)	GSE	Correlation Coefficient		
					Horizontal	Vertical	Diagonal
Proposed Method	$2^{495\;}$	99.63	33.54	7.9975	−0.0002	0.0017	−0.0027
Ref [22]	3.40 × $10^{78}$	99.56	33.44	7.9979	0.0015	−0.0014	−0.0028
Ref [41]	$2^{128}$	99.63	33.45	7.9974	0.0078	0.0040	−0.0050
Ref [13]	0.81 × $10^{192}$	99.61	33.52	7.9972	−0.0040	0.0042	0.0063
Ref [36]	$2^{144}*10^{126}$	99.61	33.45	7.9994	0.0205	−0.0128	−0.0298
Ref [42]	$2^{200}$	99.59	33.44	7.9970	0.0030	0.0029	−0.0004
Ref [44]	$2^{495}$	99.61	33.35	7.9974	0.0053	0.0011	−0.0038
Ref [46]	$10^{192}$	99.72	33.51	7.9993	0.0025	0.0019	-
Ref [45]: (Pepper Image)	$2^{3250}$	99.62	33.53	7.9993	−0.0010	−0.0292	−0.0107

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
