# Peer review of "A New Composite Fractal Function and Its Application in Image Encryption"

_2313-433X, 2020, doi:10.3390/jimaging6070070_

Round 1

Reviewer 1 Report

In this paper, the author proposes a new composite fractal function (CFF) map. The map shows high sensitivity at initial conditions, complex structure, wider chaotic regime and complex behavior according with simulation results. Furthermore, the author propose a chaos-based encryption scheme based on CFF map. Simulation results show the effectiveness of proposed scheme including security analysis and performance.

The reviewer has the following comments:

  • Some typos must be attended. g., see:
    1. Page 2 line 57: “Li e al.” must be “Li et al.”.
    2. Page 3 line 137: “MS” must be well defined.
    3. Page 3 line 142: “0.3” must be “beta=0.3”.
    4. Page 15 line 406: “&” must be “and”.
    5. Page 15 line 424: “256*256” must be “256X256”
    6. Page 19 line 507: Table 6 must show all the information.
    7. Page 22 line 579: Table 9 must show all the information.
    8. Page 24 line 686: Reference [43] must be well defined.
    9. There are some other error typos. Please review manuscript.
  • Introduction structure must be improved. In Introduction section,
    1. Appoint in a separate paragraph the main contributions of manuscript.
    2. At end of Introduction Section, the Sections of the manuscript must be described in a separate paragraph.
  • Comparisons of proposed CFF map are needed. According with author, the proposed CFF present better properties than Mandelbrot set function. Authors must compare the analysis results with MS. Please include this in Section 2.
  • Some important security analyses are omitted. Please, include important security analysis over the proposed scheme to show a more complete security analysis. Review next reference https://doi.org/10.3390/e21080815 (2019) for more details.
  • Some references are missed. Please consider the next references to support Introduction Section, security analysis and comparisons:
    1. https://doi.org/10.1016/j.jisa.2019.102390 (2019)
    2. https://doi.org/10.1016/j.ijleo.2019.02.007 (2019)
    3. DOI: 1109/ACCESS.2019.2921309 (2019)
    4. DOI: 10.1109/CompComm.2018.8780765 (2018)

Author Response

Thanks!

Reviewer 2 Report

1. The abstract needs to be enhanced with numerical results and findings. 2. Introduction first paragraph requires citation. 3. Related work must express the demerits of the earlier works. 4. The CFF is quiet good and promoted well in the manuscript. 5. Figure 8(b) is not clear. 6. What is the procedure for colour image encryption using the mentioned methodology? 7. Then and there, few of the texts are cut. Correct those mistakes. 8. Perform chosen-plaintext analysis. 9. Reference section needs to be corrected. Most of the references are incomplete. 10. Zigzag scanning is a very old technique to choose the pixel randomly (Known random). Why did not you adopt some other techniques for random pixel selection? 11. From Figure 1, the CFF response image is having a pattern. Then, how come they produce random values? Please clarify me.

Author Response

Thanks!

Round 2

Reviewer 1 Report

In this round, the authors attended all the reviewer’s comments and it has been improved. Therefore, I can recommend this paper for publication.